# Comparative Analysis of Root Phenolic Profiles and Antioxidant Activity of Five Native and Invasive *Solidago* L. Species

**DOI:** 10.3390/plants13010132

**Published:** 2024-01-02

**Authors:** Jolita Radušienė, Birutė Karpavičienė, Gabrielė Vilkickytė, Mindaugas Marksa, Lina Raudonė

**Affiliations:** 1Laboratory of Economic Botany, Nature Research Centre, Akademijos Str. 2, 08412 Vilnius, Lithuania; birute.karpaviciene@gamtc.lt; 2Laboratory of Biopharmaceutical Research, Institute of Pharmaceutical Technologies, Lithuanian University of Health Sciences, Sukileliu Av. 13, 50162 Kaunas, Lithuania; gabriele.vilkickyte@lsmu.lt (G.V.); lina.raudone@lsmu.lt (L.R.); 3Department of Analytical and Toxicological Chemistry, Lithuanian University of Health Sciences, Sukileliu Av. 13, 50162 Kaunas, Lithuania; mindaugas.marksa@lsmu.lt; 4Department of Pharmacognosy, Lithuanian University of Health Sciences, Sukileliu Av. 13, 50162 Kaunas, Lithuania

**Keywords:** alien goldenrods, phenolic acids, interspecific hybrids

## Abstract

The high environmental importance of invasive goldenrod has prompted research to find potential benefits that can be derived from these species. This study aimed to identify differences in root phenolic profiles among five *Solidago* species, thus providing valuable information on their potential applications and the botanical origin of the raw material. The roots of native *S. virgaurea* L., two alien species *S. gigantea* Aiton and *S. canadensis* L. and their hybrids *S*. ×*niederederi* Khek and *S*. ×*snarskisii* Gudž. & Žaln. were sampled from mixed-species stands in Lithuania. A complex of twelve phenolic acids and their derivatives was identified and quantified in methanol–water root extracts using the HPLC-PDA and LC/MS systems. The radical-scavenging capacities of the extracts were assessed by ABTS. The chemical content of the roots of *S. virgaurea*, *S. gigantea* and *S. ×niederederi* were statistically similar, while the roots of *S. canadensis* and *S*. ×*snarskisii* contained lower amounts of compounds than the other species. The PCA score-plot models of the phenolic profiles only partially confirmed the identification of *S*. ×*niederederi* and *S*. ×*snarskisii* as crosses between native and alien species. The findings from the phenolic profiles and the observed radical-scavenging activity of root extracts of *Solidago* species provide valuable insights into their potential applications in various fields.

## 1. Introduction

The genus *Solidago* L. (Asteraceae) is native to North America, South America and Eurasia and comprises about 139 species [1]. *Solidago* L. species, known as “goldenrods”, have a long history of cultivation and are an important genus in horticulture. *Solidago canadensis* L. (Canadian goldenrod) and *S. gigantea* Aiton (Giant goldenrod) were introduced to Europe in the mid-18th century from North America as ornamental plants, and both species later began to spread outside gardens and successfully invaded new habitats [2]. Digging and transporting soil helps to disperse rhizome fragments that can re-establish and spread to new areas [3]. In this respect, abandoned and disturbed areas contribute to rapid invasion by alien goldenrods, which sprout from rhizomes and wind-dispersed seeds. New clones increase in number via horizontal rhizomes, which form monospecific stands and eliminate surrounding species [4,5]. The ecological success of the goldenrod is due to its wide range of tolerance to different environments. Climate-change factors such as warming and nitrogen deposition are considered to facilitate the spread of goldenrods [6,7]. Allelopathy has also often been implicated in the potential of *Solidago* spp. for invasiveness and naturalization. The “novel weapons hypothesis” suggests that invasive plants have a competitive advantage over native plant species due to their allelochemicals [8].

Furthermore, the spontaneous hybridization of two North American species, *S. canadensis* and *S. gigantea*, with the native *S. virgaurea* L. (European goldenrod) produced new hybrids. *Solidago ×niederederi* Khek. a well-documented hybrid of *S. canadensis* and *S. virgaurea*, has become a common species in many European countries and is a strong competitor to its parent species, achieving similar biomass and producing rhizomes like *S. canadensis* [9,10,11]. *Solidago ×niederederi* has begun to develop its own geographic range, which seems to be closely related to the overlap in the ranges of its parent species [12]. *Solidago ×snarskisii* Gudžinskas and Žalneravičius, a hybrid between *S. virgaurea* and *S. gigantea*, was recorded for the first time in Southern Lithuania [13] and was later found in Poland, Russia and Sweden [14,15]. The emergence of new spontaneous hybrids reflects the ongoing invasion of *S. canadensis* and *S. gigantea* and their increasing impact on the native flora. According to the concept proposed by Pyšek et al. [16], the hybrids are treated as new alien taxa that spread along with the parental species. *Solidago canadensis* and *S. gigantea* were listed on the EPPO Invasive Alien Plant List [17], which lists the plants that have been identified as posing a significant threat to plant health, the environment and biodiversity. Recommendations on measures to prevent the further spread of alien species were also provided. However, goldenrods are not easily controlled by mowing or using herbicides, as the plants regrow several months after application [18]. Therefore, the problem of goldenrod control requires a multifaceted approach. Consequently, the high environmental importance of invasive goldenrods has prompted research to find potential benefits of and applications for these plants.

Goldenrod herbal material has been used in European phytotherapy for a very long time for the treatment of urinary and genital diseases [19]. *Solidaginis herba*, usually consisting of aerial parts of *S. virgaurea*, *S. canadensis* and *S. gigantea*, is included in ESCOP [20]. Many published studies describe potential activities of *S. virgaurea* plant extracts, including antioxidant, anti-inflammatory, analgesic, antispasmodic, antihypertensive, diuretic, antibacterial, antifungal, antiparasitic, cytotoxic and antitumor, antimutagenic, antiadipogenic, antidiabetic, cardioprotective and antiaging effects [21]. Furthermore, alien *Solidago* spp. have been intensively studied for their content of chemical compounds and their potential uses. A number of specialised metabolites, including caffeoylquinic acid derivatives, flavonoids, monoterpenes and sesquiterpenes, polyacetylenes, clerodane-type diterpenes, saponins and tannins have been reported in the raw materials of *Solidago* spp. [22,23,24,25,26,27,28,29,30]. The corresponding specialised metabolites have been considered as sources of functional ingredients for agriculture, food, cosmetics and pharmaceuticals, in which applications phenolic compounds and essential oils are of great importance. Specifically, caffeoylquinic acids have great potential for use in drug development due to their reported antioxidant [31,32,33], antibacterial [31,32], anticancer [34], anti-Alzheimer’s [35] and neuroprotective [36,37,38] properties.

Data regarding the chemical composition and biological activities of *Solidago* spp. roots extracts are scarce. *Solidago canadensis* root essential oil is dominated by thymol, α-copene and carvacrol. These components were respectively found to exhibit significant antibacterial activity against *Enterococcus faecalis* and *Escherichia coli* and moderate antifungal activity against *Candida albicans* [39]. Clerodane-type diterpenes were isolated from *S. canadensis* [40] and *S. gigantea* root extracts [27]. Two antimicrobial labdane diterpenes (solidagenone and presolidagenone) [41] and (-)-hardwickiic and (-)-abietic acids [29] were detected and isolated from *S. canadensis* and *S. rugosa* root extracts, respectively. In addition, matricaria ester-type polyacetylenes were identified as active major compounds in the roots of *S. canadensis* [40] and *S. virgaurea* [41]. The labdane diterpenes from the rhizomes of *S. chilensis* had a gastroprotective effect that was explained by their ability to prevent oxidative stress [42]. The antioxidant profiles of leaves and inflorescences of invasive and native *Solidago* spp. were determined by extracting major markers with antioxidant properties [43].

Continuing the research on the bioactive compounds of invasive *Solidago* spp., this study aimed to characterise the phenolic compound profiles of five *Solidago* spp. root extracts, describe their antioxidant activities and provide a comparative analysis. The results of the previous study and the present study allowed us to verify the following statements: (i) *Solidago* species and populations differ in their concentrations of phenolic metabolites in root extracts and in the antioxidant activity of these metabolites; (ii) the concentration of phenolic compounds in the roots/rhizomes of invasive *Solidago* spp. is expected to be higher compared to that in the roots of native species; (iii) *Solidago* spp. growing in mixed-species stands provide a more robust comparison because it removes the effect of different environments on species differences; (iv) the origin of interspecific hybrids can be confirmed by comparing the chemical composition of their roots to the chemical composition of the roots of the parental species. To the best of our knowledge, this study is the first presentation of a chemical comparison used to identify and quantify phenolic compounds and their antioxidant activities in five *Solidago* spp. root extracts.

## 2. Results

### 2.1. Phenolic Profiling and Comparative Quantitative of Solidago spp. Roots

The results of the root fingerprint analysis revealed differences in the complex of phenolic acids and their derivatives among *Solidago* spp. A total of twelve phenolic acids and their derivatives were identified and quantified in the roots of native *S. virgaurea,* two alien species, *S. gigantea* and *S. canadensis*, and two spontaneous hybrids, *S. ×niederederi* and *S. ×snarskisii* (Figure 1, Table 1).

The quantitative results revealed that the mean concentrations of individual phenolic acids and their derivatives differ significantly (*p* < 0.05) between species. Furthermore, the results of the ANOVA and non-parametric multiple comparisons by the Kruskal–Wallis test, presented in Table 2, indicate similarities and differences in the concentrations of individual phenolic compounds among the roots of five *Solidago* spp.

The root extracts of all species were dominated by a tentatively identified dicaffeoylquinic acid derivative, the mean concentration of which ranged from 5094.1 to 13,666.3 µg/g DW. By a significant margin (*p* < 0.05), the greatest concentration of dicaffeoylquinic acid derivative was detected in the roots of *S. virgaurea*, followed by the roots of *S.* ×*niederederi* and *S. gigantea*. Moreover, high levels of derivatives of phenolic acid I, II, III and IV, together with chlorogenic acid, were detected in roots, with clear differences between species. The greatest amounts of phenolic acid I and III derivatives were found in the root extracts of *S*. ×*niederederi* (4052.3 and 3620.2 µg/g DW, respectively), while a phenolic acid IV derivative was present at its highest concentration in *S. virgaurea* root extracts (6606.52 µg/g DW). The level of phenolic acid II derivative did not differ significantly between four of the species, but it was present in a lower amount in *S. canadensis*.

In another significant finding, the greatest amount of chlorogenic acid was found in the roots of *S. gigantea* (6360.9 µg/g DW), followed in decreasing order by *S*. ×*snarskisii*, *S. virgaurea*, *S*. ×*niederederi* and *S. canadensis*. In addition, *Solidago* spp. roots contained small amounts of other mono-caffeoylquinic acids as neochlorogenic and cryptochlorogenic acids, together with 1,5-O-dicaffeoylquinic acid. The highest levels of the caffeoylquinic acid derivatives, namely 3,5-dicaffeoylquinic acid, 4,5-dicaffeoylquinic acid and 3,4-dicaffeoylquinic acid (433.5, 2836.7, 2197.3 and 2354.6 µg/g DW, respectively) were detected in the roots of *S. gigantea*. Meanwhile, *S. canadensis* was present in the lowest concentrations among all phenolic acids and their derivatives, except for the phenolic acid III derivative, which was present in higher amounts. The results of the multiple-comparisons test indicated that the total levels of phenolic acids and their derivatives detected in the roots of *S. virgaurea*, *S. gigantea* and *S*. ×*niederederi* were statistically similar (*p* > 0.05). The total amounts of phenolic acids were lower in the roots of *S. canadensis* and *S*. ×*snarskisii* than in the other three species.

Overall, the results of this study showed that the roots of different *Solidago* spp. synthesized the same phenolic compounds, but with pronounced differences in the amounts of individual compounds between species. In the comparison of quantities of phenolic acids and their derivatives in root extracts, the native *S. virgaurea* was found to contain the highest levels of neochlorogenic acid and tentatively identified dicaffeoylquinic acid and phenolic acid IV derivatives. *Solidago* ×*niederederi* exceeded the parental and other invasive species in mean quantities of phenolic acid I and III derivatives. Roots of *S. gigantea* contained the highest concentrations of chlorogenic acid, followed by its hybrid *S*. ×*snarskisii*. *S. canadensis* contain the lowest or among the lowest amounts of all detected compounds among the tested species.

### 2.2. Intraspecific Chemical Differences

Intraspecific differences in the amounts of individual phenolic acids and their derivatives were found in *Solidago* spp. populations. The native *S. virgaurea* was associated with the greatest variation in the amounts of chlorogenic acid, 1,5-dicaffeoylquinic acid, dicaffeoylquinic acid derivative and phenolic acid III and IV derivatives (Table 2). Populations of the invasive *S. canadensis* and *S. gigantea* differed in the concentrations of dicaffeoylquinic acid derivative. Furthermore, populations of *S. canadensis* and both spontaneous hybrids, *S*. ×*niederederi* and *S*. ×*snarskisii*, showed intraspecific differences in levels of neochlorogenic acid. However, no intraspecific differences in the concentrations of 4,5-, 3,5- and 3,4-dicaffeoylquinic acids were found in the roots of the studied species. Differences in total phenolic acids and their derivatives were found between native and alien populations of *Solidago* spp., but not in their hybrid populations.

### 2.3. Principal Component Analysis: Relationships between Species

We evaluated the possibility of using root phenolic acids and their derivatives to identify the origins of interspecific hybrids. Principal component analysis (PCA) offered a general overview of the variance in concentrations of specialized metabolites and was performed to visualise the phytochemical relationships between the *Solidago* spp. Individual phenolic acids and their derivatives were used as independent variables in two-dimensional PCA1 and PCA2 scatter plots. The results of the correlation matrices for the variables and the principal components in the PCA1 and PCA2 models are presented in Table 3.

The PCA1 model explained more than 62% of the total variance in the data set and was specifically focused on the location of the *S. virgaurea*, *S. canadensis* and *S*. ×*niederederi* root samples (Figure 2). PC1 accounted for 45.33% of the total variance of the data set and was highly positively correlated with all variables except for the concentration of the phenolic acid III derivative (Table 3). Variables with high loadings on PC1 contributed the highest impact to the grouping of *S. virgaurea* samples, which were widely spread along PC1, displaying great variation in chemical composition (Figure 2a). PC2 explained 16.81% of the total variance in the data set and was highly correlated with the positive loadings of phenolic acid I and III derivatives, which had the greatest influence on the placement of *S*. ×*niederederi* samples. Most of the *S. canadensis* samples were clustered close to PC1 and PC2 zero, indicating a range of variable values from the lowest to the mean (Figure 2b). Meanwhile, the positions of some *S*. ×*niederederi* samples partially overlapped with those of *S. canadensis* samples, indicating similarity in their chemical contents, while some other samples clustered near to *S. virgaurea*. In this way, the locations of *S*. ×*niederederi* samples close to the parental species indicated their similarity to both species in terms of values of the relevant variables. In addition, some *S.* ×*niederederi* samples spread outside the range of parental scores on the scatterplot, indicating that they contain greater diversity in their values than the parental species have.

The PCA2 model accounted for 65% of the total variance in the data set and was used to explain the locations of the *S. virgaurea*, *S*. ×*snarskisii* and *S. gigantea* samples (Figure 3). The PC1 correlation matrix described 38.83% of the total variance in the data set and showed a positive correlation with the concentrations of cryptochlorogenic and chlorogenic acids; 4,5-, 3,5- and 3,4-dicaffeoylquinic acids; dicaffeoylquinic acid derivative and phenolic acid I and II derivatives (Table 3). PC2 explained 26.12% of the total variance and was highly associated with the positive loadings of chlorogenic acid and 3,5- 3,4-dicaffeoylquinic acids and negative loadings of dicaffeoylquinic acid derivative and phenolic acid III and IV derivatives (Figure 3a). The locations of *S. gigantea* samples in a somewhat isolated position on the upper half of the score plot space can be explained by the contribution of variables with high loadings on both PC1 and PC2 (Figure 3b). The clustering of *S. gigantea* samples was associated with high loadings of chlorogenic acid and 3,5-, 3,4- and 4,5-dicaffeoylquinic acids, which were considered to be the principal compounds of this species. Meanwhile, the grouping of most *S. virgaurea* samples was greatly influenced by the dicaffeoylquinic acid derivative and the phenolic acid III and IV derivatives.

The variables with high loadings on PC2 had a significant impact on the arrangement of *S*. ×*snarskisii* scores; these samples were located in an intermediate position between the *S. virgaurea* and *S. gigantea* samples (Figure 3b). On the other hand, the small number of *S*. ×*snarskisii* samples did not allow us to determine the position of the hybrid in the score plot in relation to the parental species with high certainty. Scattering of root samples in the PC1 and PC2 score plot spaces confirmed, as in the previous model, a high level of variability in the concentrations of chemicals in *Solidago* spp. roots.

This PCA of the phenolic profiles of *Solidago* spp. roots did not reveal well-defined clusters of samples with species-specific compounds, indicating that these plants have high intraspecific chemical diversity. This finding is especially notable for *S. virgaurea* and *S*. ×*niederederi*. Thus, PCA models only partially confirmed the origins of the spontaneous hybrids *S*. ×*niederederi* and *S*. ×*snarskisii* as crosses between native *S. virgaurea* and invasive *S. canadensis* and *S. gigantea*, respectively.

### 2.4. Antioxidant Activity

Antioxidant profiles of *Solidago* spp. root extracts composed of mono- and dicaffeoylquinic acids and their derivatives revealed some significant differences (*p* ≤ 0.05) between species. The root extracts of *S. virgaurea* and *S. ×niederederi* yielded the highest mean values of radical-scavenging activity (205.0 and 193.1 µmol TE/g, respectively), followed by *S. gigantea*, *S. canadensis* and *S. ×snarskisii* (187.4, 154.4 and 131.9 µmol TE/g, respectively) (Figure 4). The antioxidant activity of native *S. virgaurea* was significantly higher than that of the invasive species *S. canadensis* and *S*. ×*snarskisii*. The difference between the *S. virgaurea* and *S. gigantea* and *S*. ×*niederederi* was statistically insignificant. Therefore, it can be assumed that antioxidant activity is not related to the invasiveness of the studied species.

However, the free-radical-scavenging capacity of *Solidago* spp. root extracts did not show intraspecific differences among populations, except for marked differences (*p* < 0.05) among *S. gigantea* populations (Table 2). In this way, phenolic profiles of *S gigantea* roots exhibited significantly more variation in radical-scavenging power among populations than did other *Solidago* spp. According to a Pearson’s correlation analysis, the antioxidant activity of root extracts was correlated with the concentrations of individual and total phenolic compounds (Table 4).

The radical-scavenging activity showed strong positive correlations (*p* < 0.001) with the amounts of dicaffeoylquinic acid derivative (r = 0.63), phenolic acid II derivative (r = 0.57), cryptochlorogenic acid (r = 0.51), phenolic acid I derivative (r = 0.49), neochlorogenic acid (r = 0.43), and 4,5-dicaffeoylquinic acid (r = 0.40). Thus, it can be concluded that the profiles of phenolic acids and their derivatives in *Solidago* spp. root extracts were potentially responsible for the detected antioxidant activity, which depended on the individual compounds and their concentrations. According to this analysis, dicaffeoylquinic acid, together with phenolic acid I, II and III derivatives, which were predominant in the root extracts of *S. virgaurea* and *S. ×niederederi*, apparently contributed the most to their antioxidant activity. Thus, the phenolic profiles of different *Solidago* spp. roots have different antioxidant capacities, and these capacities were strongly correlated with the amounts of phenolic acids and their derivatives in the root materials. The results are in agreement with the results of previous studies, which reported a strong correlation between content of phenolics and antioxidant activity for different plant extracts [44]. Furthermore, each phenolic compound reacted differently, and antioxidant potency is closely related to molecular structure [45]. As in our identification, previous ABTS post-column studies indicated that the highest radical-scavenging activity in the leaves and inflorescences of *Solidago* spp. was detected for a complex of caffeoylquinic acid derivatives [43].

## 3. Discussion

Despite the importance of goldenrods’ effects on the surrounding environment and uses in herbal medicine, very little research has been done on the potential uses of the roots/rhizomes, which have never been used for commercial purposes. Additionally, studies on the underground organs of *Solidago* spp. are important in other respects, for example, in the control of invasion, while the possible uses of the goldenrod roots may have direct implications, with root harvesting as an approach by which to reduce their spread. Phenolic compounds were considered to contribute to the invasive potential of alien species by influencing their interactions with other organisms, modifying the physical and chemical characteristics of the rhizosphere and thus enhancing the competitive ability of the invader [7,8,46]. On the other hand, phenolic compounds are also known to have apparent health benefits and great potential for use in drug development due to their anti-fungal, anti-bacterial, antioxidant and anti-inflammatory activities, as well as their immunomodulatory, anticancer and neuroprotective properties. Phenolic compounds are considered promising agents against human viruses as well [47].

Our hypothesis was based on previous observations of the aerial parts of the plant, which suggested that the roots of the invasive goldenrod, like the aerial parts, accumulate greater amounts of phenolic compounds than those of the native species. Thus, previous data on phenolic compounds in leaves and inflorescences of native and alien *Solidago* spp. were compared with the data obtained here [30,43,48]. However, the findings were surprising, as the roots of *S. virgaurea* accumulated high or the highest levels of individual phenolic acids and their derivatives compared to the invasive *Solidago* spp. Meanwhile, the roots of the aggressive invader *S. canadensis* were characterised by low or the lowest mean values of phenolic acid concentrations. On the other hand, *S. canadensis* leaves and inflorescences, compared to those of *S. ×niederederi* and *S. virgaurea*, contained higher levels of phenolic compounds. The findings from the roots of *S. gigantea* were in agreement with the high values of phenolic compounds in the leaves and inflorescences compared to other invasive goldenrods [48]. The most common cytotype of *S. gigantea* is tetraploid [49], while the other species are diploid; this cytotype is associated with the highest levels of phenolic compounds of all plant parts studied. Ploidy is known to have significant effects on the accumulation of secondary metabolites [50]. Higher levels of phenolic compounds were detected in the rhizomes/roots of invasive *Reynoutria japonica* Houtt. compared to native plant species [51]. However, such a comparison does not explain our results because the accompanying resident species were not individually identified. There are no data available that can be used to compare the phenolic profiles of *Solidago* spp. roots with the results of studies conducted in other locations.

Moreover, differences in chemical profiles between species can also be related to differences in the structures of the root systems of the studied species (Figure 5). The hybrids, at least in the F1 generation, did not produce long rhizomes, as were typical of both the alien *S. canadensis* and *S. gigantea*. *S. ×niederederi* and *S. ×snarskisii* formed short rhizomes with a large clump of roots similar to that of native *S. virgaurea*, which formed thick crowns with multiple roots. Nevertheless, the nature of the relationship between the structures of the root systems and their chemical compositions is not obvious, so further research on root morphology and chemistry has potential elucidating both the invasion processes and the potential uses of the raw materials. Qualitative and quantitative differences in the complex of phenolic compounds were observed between plant roots and the previously assessed aerial parts of the plant. Using the same analytical procedures, no quercetin derivatives were detected in the root extracts. Those compounds, together with caffeoylquinic acids, dominated in the phenolic profiles of the leaves and inflorescences of the corresponding *Solidago* spp. [43]. On the other hand, high levels of newly tentatively identified phenolic acids and their derivatives were detected in *Solidago* spp. root profiles. These findings suggest that the roots and aerial parts of *Solidago* spp. exhibit different species-specific patterns in their concentrations of phenolic compounds. The choice of plant organ is very important in the evaluation and selection of raw materials from plants and in gaining further insights into their potential applications.

*Solidago virgaurea* has been described as a highly polymorphic species combining closely related taxa at different levels [52,53]. Our study confirmed that the previously described taxonomic polymorphism of *S. virgaurea* was accompanied by a high intraspecific diversity in the concentrations of phenolic acids and their derivatives in the roots. This diversity apparently increased its rate of survival in mixed stands with alien invaders. On the other hand, *S. virgaurea* was considered to be a taxon with low competitive ability that is commonly displaced from its habitat by invasive goldenrods [54]. In addition, it was found that the allelopathic activity of *S. virgaurea* root and inflorescence extracts was higher than those of *S. canadensis* and *S. ×niederederi* [55]. This activity likely improves its survival in mixed-species stands as well. Furthermore, the finding that the antioxidant capacity of *S. virgaurea* extracts was higher than that of invasive species can be considered to indicate its greater potential to adapt to invaders [56]. On the other hand, the opposite trend was observed in leaves and inflorescences, as the antioxidant capacities of these organs in the invaders were significantly higher than those of the native species [43].

Overall, the differences in antioxidant capacity confirmed that phenolic compounds contribute to the differences in antioxidant capacity in different goldenrod organs. Such differences in scavenging activity were reported even in different parts of the *Pyrus communis* L. fruit, such as the exocarp, endocarp and hypanthium, and depended on the concentrations of the main compounds in these parts [57]. Thus, the quantification of the individual phenolics is important for predicting the potential antioxidant capacity of different parts of the plant and facilitating a more targeted use of plant materials. In this way, the high concentrations of dicaffeoylquinic acid and phenolic acid I, II and IV derivatives apparently resulted in *S. virgaurea* having the antioxidant activity among tested species. Chlorogenic acid, as the main phenolic compound found in the phenolic profiles of both aerial parts and roots of *S. gigantea*, can be considered a marker of the antioxidant activity of this species.

In this context, it was important to use the same method of assessing radical-scavenging capacity to compare the relative antioxidant activities of different plant materials and assess the trends in their variation. Thus, the ABTS assay, which is a well-known method for the determination of total antioxidant activity and is applicable to both lipophilic and hydrophilic compounds, was used as in previous studies of *Solidago* spp. to assess radical scavenging by phenolic compounds [43].

The invasion success of goldenrod is thought to be due to allelochemicals produced in the roots and aerial parts of the native species that can inhibit the growth of other species. It was reported that during invasion, *S. canadensis* significantly increases the synthesis of specialised metabolites with high allelopathic potential. These metabolites enter the rhizosphere as root exudates or during the decomposition of plant residues [58,59,60]. Total flavones, total phenolics and total saponins were considered to be the major allelochemicals whose concentrations in the rhizosphere soil of *S. canadensis* were higher in invaded ranges than in native ones [58,61]. However, there are no data on the specific individual root compounds whose production increases in the invasive ranges. Past studies have focused mainly on the evaluation of *S. canadensis*. Aqueous ethanolic extracts of the roots and rhizomes of *S. canadensis* inhibited the germination and growth of different plant species and the mycorrhizal fungal population in its rhizosphere soil [60]. Meanwhile, the root essential oil, which was dominated by limonene and *β*-pinene, suppressed the germination of lettuce seeds and showed significant inhibitory effects on the growth of lettuce and garden pepper cress [55]. In this way, the effects of the extracts on other species make them potentially useful as bioherbicides that can be used to help manage a range of pest and disease problems.

Species hybridization due to transgressive segregation may facilitate adaptation and may be associated with an increase in phenotypic diversity outside the parental range. This diversity may be particularly important in extreme environments, where the emergence and spread of hybrids may increase threats to native species [62]. On the other hand, we studied hybrids that occur only in close sympatry with the parental species, in mixed-species stands. Our hypothesis was based on previous results that showed an obvious contribution of the parental species to the chemical composition of the aerial parts of *S. ×niederederi*; however, the presented root phenolic profiles could be only partially explained by the effect of parental species on the quantitative chemical composition of the hybrids. The high chemical diversity of hybrid populations can be attributed to the increased access to mutations and genotypes that results from their origin from two different genomes [63]. Therefore, hybrids may have an adaptive advantage over other species when faced with stressful conditions. They may thus also have a greater potential for spread*. Solidago ×niederederi* was found to produce viable seeds [10,64], while no fruit development has been observed in *S. ×snarskisii* so far, indicating its sterility [14]. For this reason, *S. ×niederederi* is treated as an established alien, while *S. ×snarskisii* is currently treated as an accidental species [65]. Thus, the spread of *S. ×snarskisii* and the source of its raw material are currently considered to be of little importance.

Plant hybrids inherit most of their chemicals have in a Mendelian pattern of dominance. Thus, if one or both parental species produce a chemical, the hybrids will almost always synthesise the same chemical, usually either at a concentration similar to that produced by one of the parents or at an intermediate level [66,67]. In of our study, the mean concentrations of individual phenolic acids and their derivatives in the roots of *S. ×niederederi* were at levels similar to those seen in one of the parental species, *S. virgaurea* or *S. canadensis*. Meanwhile, the mean concentrations of most of the individual compounds present in the roots of *S. ×snarskisii* were expressed at intermediate levels with respect to the parental species. Nevertheless, the multivariate comparative analysis found that root phenolic profiles were only partially explained by the impact of parental species on the root chemical composition of interspecific hybrids. The findings of the study provide limited evidence for the origin of *S. ×niederederi* and *S. ×snarskisii*, suggesting that other, less diverse specialised metabolites than phenolic acids and their derivatives may provide better demonstrations of the chemical consequences of hybridization between native and alien species.

It should be noted that in this study, we attempted to eliminate the effects of soil physicochemical characteristics on species diversity and so assume that this variable had no significant effect on our data, especially as the roots were collected from mixed-species stands in the same area. Moreover, in our previous study, we did not find a significant relationship between morphological features of the species and the soil composition in their growing sites [64]. The same trend, that high phenotypic plasticity was not related to substrate differences, was observed by Szymura et al. [53]. Therefore, we considered that the diversification of *Solidago* spp. phenolic profiles was related not to the plasticity of the species under different conditions, but to phytochemical heterogeneity among clones. Consequently, the phytochemical heterogeneity of *Solidago* spp. has a significant impact on the chemical composition and antioxidant capacity of root extracts and poses problems for the primary collection of valuable raw materials and subsequent standardization of final plant products.

## 4. Material and Methods

### 4.1. Plant Material and Sampling

The underground organs of *Solidago* spp. consisted of nodes and rhizomes. The structural and morphological characteristics of these organs varied between species (Figure 5).

The root/rhizomes system of *S. virgaurea* consisted of a branched underground caudex and woody cylindrical rhizomes with root clumps growing from the bottom part. *Solidago canadensis* had short-branched rhizomes that originated from nodes at the base of the stem. The rhizomes of *S. canadensis* produced compact individual clones that were easily identified by the presence of shoot clumps. The rhizomes of *S. gigantea* were long and creeping and broke easily at the nodes because the internodes were only weakly attached to each other. The rhizomes of the individual clones were tangled, making them difficult to separate. *Solidago ×niederederi* and *S. ×snarskisii* formed woody, short rhizomes with a large clump of roots.

The root/rhizomes (hereafter, roots) of five *Solidago* spp. were sampled from mixed-species stands in nineteen different sites in Lithuania in October 2022 (Figure 6).

Within each site, three root accessions of the same species were excavated from different clusters at least five meters apart to ensure that individual genets were sampled. Forty-two *S. virgaurea*, forty-four *S. canadensis*, thirty-nine *S. ×niederederi*, twenty-one *S. gigantea* and four *S. ×snarskisii* root accessions were collected from different sites (Table A1, Appendix A). The sampled roots were thoroughly washed under running water to remove soil and then dried in a drying cupboard at 50 °C for 24 h. The dried roots were stored in a refrigerator until chemical analysis.

The vegetation of the collection sites was characterised as semi-ruderal dry or mesic grassland dominated by plant communities of *Agropyretea intermedia-repentis* (Oberd. et al. 1967) Müller et Görs 1969 and *Artemisietea vulgaris* Lohm., Prsg et R. Tx. in R.Tx. 1950. One sample of *S. ×snarskisii* was taken from a field plant collection, as this hybrid is rare in nature.

The botanical identification of species was carried out by B. Karpavičienė using diagnostic morphological characters described in previous studies [13,63]. *Solidago* taxa were identified according to stem colour and hairiness, the shape and size of the inflorescence and characteristics of the capitulum and florets such as the size and number of tubular and ligular florets, the ratio of tubular to ligular florets in the capitulum and the length of the involucre. The characters of florets were measured on fresh material in the laboratory using a binocular microscope.

The specimens of *Solidago* species were deposited in the Herbarium of the Institute of Botany of Nature Research Centre, Vilnius.

### 4.2. Chemicals and Reagents

HPLC-grade standard substances, namely neochlorogenic acid (5-caffeoylquinic acid), chlorogenic acid (3-caffeoylquinic acid), cryptochlorogenic acid (4-dicaffeoylquinic acid), 4,5-dicaffeoylquinic acid, 1,5-dicaffeoylquinic acid, 3,5-dicaffeoylquinic acid and 3,4-dicaffeoylquinic acid were purchased from Sigma–Aldrich GmbH (Steinheim, Germany). Solvents and other substances were obtained as follows: acetonitrile (purity 99.9%) and methanol (purity 99.9%) from Sigma-Aldrich (Steinheim, Germany); trifluoroacetic acid (purity 99.8%) supplied from Merck (Darmstadt, Germany); 2,2’-Azino-bis(3-ethylbenzothiazoline-6-sulfonic acid) di-ammonium salt (ABTS) (purity ≥ 98%) and potassium persulphate (purity ≥ 99%) purchased from Sigma (St. Louis, MO, USA); Trolox (purity ≥ 98%) received from Fluka Chemika (Buchs, Switzerland). Ultrapure water was purified using a Millipore water-cleaning system (Bedford, MA, USA).

### 4.3. Extraction

The air-dried roots were mechanically ground up with a laboratory mill to form a homogenous powder. Samples of approximately 0.1 g (accuracy of 0.0001 g) were extracted in 10 mL of a mixture of methanol and water (70:30 *v/v*) for 30 min at 40 °C in an ultrasonic bath Elmasonic P (Singen, Germany). The extracts were filtered through 0.22 mm pore-size membrane filters (Carl Roth GmbH & Co. KG, Karlsruhe, Germany) and stored in the refrigerator at 4 °C until analysis.

### 4.4. HPLC—PDA Analysis

The HPLC analysis was performed using a Waters Alliance 2695 mode system coupled with a 2996 PDA photodiode-array detector (Waters, Milford, MA, USA). The phenolic compounds were separated using an ACE Super C18 (250 mm × 4.6 mm i.d., 3.0 µm) column (ACT, Aberdeen, UK) maintained at 35 °C. The mobile phase of binary gradient elution at a flow rate of 0.5 mL/min consisted of eluent A (0.1% trifluoroacetic acid in pure water) and eluent B (100% acetonitrile). The elution program was as follows: 0–40 min—10–30% B, 40–60 min—30–70% B, 60–64 min—70–90% B, and 64–70 min—90–10% B. The injection volume of the extract was 10 µL.

The analysis was performed using full scan on the interval of 200–400 nm. Identification was performed by scanning in a range of 200–400 nm wavelengths by comparing UV/Vis spectral data and retention times to those of standard compounds (Figure 1). The data on the validation method were provided in previous papers [68,69].

### 4.5. HPLC—MS Analysis

The LC/MS system was composed of a Shimadzu Nexera X2 LC-30AD HPLC system (Shimadzu, Tokyo, Japan) equipped with a LCMS-2020 mass spectrometer (Shimadzu, Tokyo, Japan). An ACE Super C18 (250 mm × 4.6 mm i.d., 3.0 µm) column (ACT, Aberdeen, UK) was maintained at 35 °C. The mobile phase of the binary gradient elution at a flow rate of 0.5 mL/min consisted of eluent A (0.1% trifluoroacetic acid in water) and eluent B (100% acetonitrile). The elution program was as follows: 0–40 min—10–30% B, 40–60 min—30–70% B, 60–64 min—70–90% B, and 64–70 min—90–10% B. The injection volume of extract was 5 µL. The flow rate was 0.5 mL/min; the column temperature was 35 °C. HPLC was run at 0.5 mL/min flow. The optimum electrospray ionization (ESI) mode conditions were set at 350 °C for interface temperature, 250 °C for DL temperature, 400 °C for heat-block temperature, 1.5 L/min for nebulising gas flow, and 10 L/min for drying gas flow. Positive and negative ion measurements were performed while switching alternately between positive and negative ionization modes. MS spectra were recorded in a range of 50 to 2000 at scan speed 15,000 µ/s, with 0.2 *m*/*z* steps.

### 4.6. ABTS Radical-Scavenging Assay

The radical-scavenging capacity of root extracts was analysed by radical-cation decolorization assay (ABTS) based on the ability to scavenge ABTS^•+^ radicals according to previously outlined and modified protocols [69,70]. The results were expressed as micromolar Trolox equivalent per gram dry weight of extract (μM TE/g DW).

### 4.7. Statistical Analysis

Statistical analysis of the data was performed using the Statistica 10.0 software package (StatSoft Inc., Hamburg, Germany). The non-parametric Kruskal–Wallis test was used to determine differences between species. Significant differences among populations of each species were tested using ANOVA. Principal component analysis (PCA) was used to identify similarities and differences between samples analysed using statistically independent variables. As the data were not normally distributed, a logarithmic transformation was applied prior to PCA analysis. After transformation, all 12 standardised variables were used in the PCA. Pearson’s correlation analysis was performed to elucidate the relationship between content of phenolic compounds and the antioxidant capacity of root extracts.

## 5. Conclusions

The environmental importance of invasive *Solidago* spp. has prompted a new approach to their management that considers their potential as a source of high-added-value raw materials. Previous and current studies of the phenolic profiles of native and invasive *Solidago* species have provided different insights into the qualitative and quantitative differences between aboveground and belowground organ systems. The findings suggest that the roots and aerial parts of *Solidago* spp. exhibit specific patterns in their concentrations of phenolic compounds. For this reason, the effect of the plant organ is very important in the selection of plant raw materials and prediction of their applications. The combination of fingerprinting and multivariate data analysis allowed us to demonstrate a simplified estimation of the parentage of *S*. ×*niederederi* and *S*. ×*snarskisii*. Our data could be only partially explained by the effect of parental species on the quantitative expression of mono- and dicaffeoylquinic acids and their derivatives in hybrids. The findings suggest that other specialised metabolites that are less diverse than phenolic acids and their derivatives may be more strongly associated with the hybridization between native and alien species and its effects on chemical composition. Phytochemical heterogeneity among *Solidago* spp. greatly affects the chemical compositions and antioxidant capacities of root extracts and poses problems for the initial collection of valuable raw materials and the subsequent standardization of final plant products. Thus, the comparative analysis of phenolic profiles of goldenrod roots, here presented for the first time, showed that the roots of both native and invasive *Solidago* spp., including their spontaneous hybrids, can be treated as important sources of phenolic acids and their derivatives.

## Figures and Tables

**Figure 1 plants-13-00132-f001:**
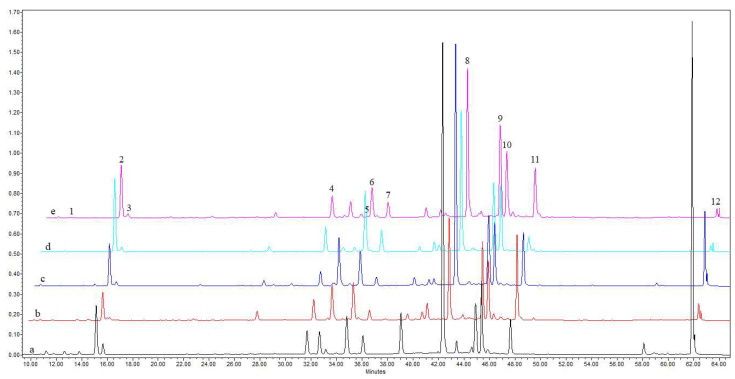
Comparison of HPLC-PDA chromatogram profiles (λ = 330 nm) of root extracts of five *Solidago* spp.: (a) *S. virgaurea*; (b) *S. canadensis*; (c) S. ×*niederederi*; (d) *S. gigantea*; (e) *S*. ×*snarskisii*. Peak assignments: 1—neochlorogenic acid, 2—chlorogenic acid, 3—cryptochlorogenic acid, 4—4,5-dicaffeoylquinic acid, 5—1,5-dicaffeoylquinic acid, 6—3,5-dicaffeoylquinic acid, 7—3,4-dicaffeoylquinic acid, 8—derivative of dicaffeoylquinic acid, 9—derivative of phenolic acid I, 10—derivative of phenolic acid II, 11—derivative of phenolic acid III, 12—derivative of phenolic acid IV.

**Figure 2 plants-13-00132-f002:**
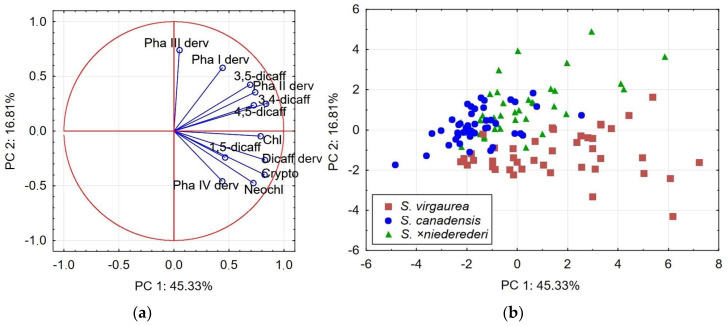
PCA1 model representing the concentrations of phenolic compounds in roots of *S. virgaurea*, *S. canadensis* and *S*. ×*niederederi*: (**a**) Loading plot of the variables contributing to PC1 and PC2; (**b**) Score plot of the testing root samples.

**Figure 3 plants-13-00132-f003:**
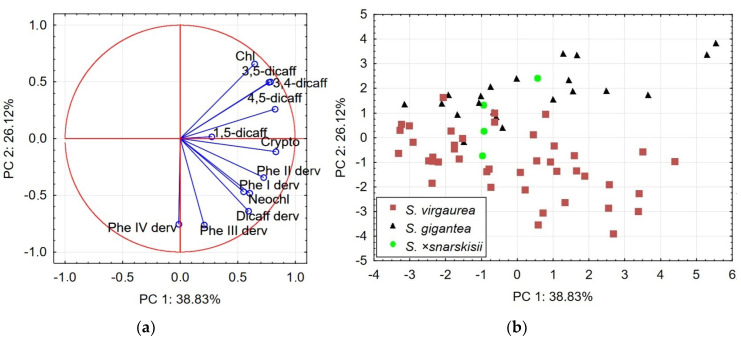
PCA2 model representing the concentrations of phenolic compounds in roots of *S. virgaurea*, *S. gigantea* and *S*. ×*snarskisii*: (**a**); Loading plot of the variables contributing to PC1 and PC2; (**b**) Score plot of the testing samples.

**Figure 4 plants-13-00132-f004:**
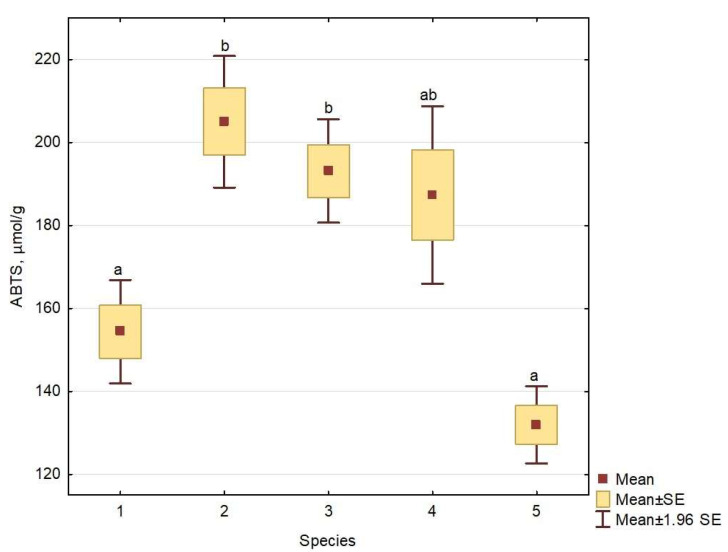
Radical-scavenging activity (ABTS) of the root extracts of *Solidago* spp.: 1—*S. virgaurea*; 2—*S. canadensis*; 3—*S. ×niederederi*; 4—*S. gigantea*; 5—*S. ×snarskisii*. Values followed by the different letters differ significantly between species according to the Kruskal–Wallis test (*p* ≤ 0.05).

**Figure 5 plants-13-00132-f005:**
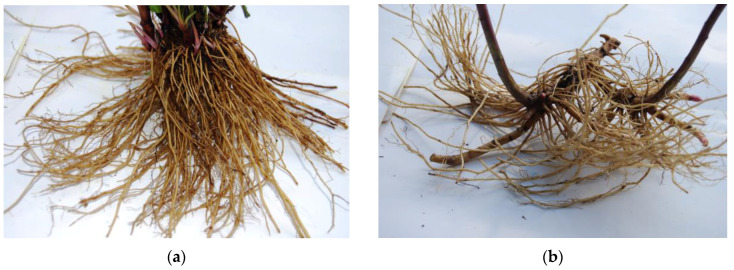
The roots of *Solidago* spp.: (**a**) *S. virgaurea*; (**b**) *S. canadensis*; (**c**) *S. ×niederederi*; (**d**) *S. gigantea* (**e**) *S. ×snarskisii*. Photos by the authors.

**Figure 6 plants-13-00132-f006:**
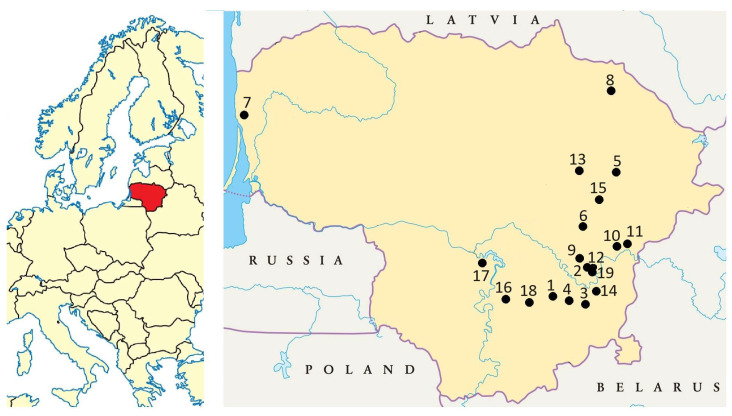
Map of five *Solidago* species sampling sites in Lithuania: Numbers indicate *Solidago* spp. sampling sites.

**Table 1 plants-13-00132-t001:** Compounds tentatively identified in extracts of different *Solidago* spp. by HPLC-MS.

Peak no.	Rt (min)	Tentative Identity	UV Max (nm)	[M-H]^−^ (*m/z*)	Other Ions (*m/z*)
8	42.3	* Derivative of dicaffeoylquinic acid	217, 245, 292 (sh), 325	629	585, 543, 113
9	44.9	Derivative of phenolic acid I (Leiocarposide isomer)	218, 292 (sh), 314 (sh), 325	613	569
10	45.3	Derivative of phenolic acid II (Leiocarposide isomer)	218, 292 (sh), 314 (sh), 325	613	569
11	47.6	Derivative of phenolic acid III (Leiocarposide isomer)	217, 240, 292 (sh), 316 (sh), 325	613	569
12	61.9	Derivative of phenolic acid IV (Methyl 3,5-dicaffeoylquinate)	209, 245, 311, 325 (sh)	361	113

* Tentatively identified compounds were quantified based on the calibration curve of chlorogenic acid.

**Table 2 plants-13-00132-t002:** Mean compound content (µg/g, DM), range of concentrations and total antioxidant capacity (TE, µmol/g) of *Solidago* spp. root extracts, with differences in compound concentrations compared between species by a Kruskal–Wallis multiple-comparisons test and between populations by ANOVA.

Compounds	*S. virgaurea* (42) ^1^		*S. candensis* (44)		*S. ×niederederi* (39)		*S. gigantea* (21)		*S. ×snarskisii* (4)	
	M (±SD)	Range	M (±SD)	Range	M (±SD)	Range	M (±SD)	Range	M (±SD)	Range
Neochlorogenic acid	312.3 b (244.7)	94.3–1108.9	88.6 a ^2^ (38.4)	61.3–301.2 *	106.0 ac (33.8)	64.0–212.1 **	141.7 c (93.2)	80.2–448.8	112.7 abc (19.5)	83.9–125.5 *
Chlorogenic acid	3181.3 b (1428.0)	731.8–6229.2 *	1827.7 a (1190.5)	696.2–8509.0	2273.0 ac (1421.0)	696.5–6526.4 *	6360.9 d (2935.1)	2566.8–13,252.0	4118.3 bcd (1611.5)	2748.7–6447.0
Cryptochlorogenic acid	452.4 b (282.0)	100.7–1185.5	120.7 a (46.5)	44.0–240.5	180.0 a (95.1)	81.7–515.7	433.5 b (302.8)	171.0–1449.1	249.0 b (52.8)	214.8–327.3
4,5-dicaffeoylquinic acid	1752.4 ab (1212.7)	201.6–4826.7	1371.0 a (1104.2)	180.7–5955.6	1683.8 ab (1018.1)	464.3–4716.7	2354.6 b (1401.2)	495.1–5688.3	1723.7 ab (313.3)	1483.2–2182.0
1,5-dicaffeoylquinic acid	69.7 b (29.2)	19.7–145.9 *	43.6 a (17.9)	22.0–95.4	49.4 a (28.7)	26.8–170.8	72.1 b (24.7)	46.8–135.6	86.5 b (14.1)	68.6–103.2
3,5-dicaffeoylquinic acid	1622.3 ab (942.0)	351.0–3988.0	1229.6 a (649.0)	412.7–3584.6	2064.4 bc (1469.1)	312.5–7472.1	2836.7 c (1310.0)	1094.5–5604.4	2160.6 abc (1078.5)	1033.0–3623.5
3,4-dicaffeoylquinic acid	950.4 a (630.5)	116.4–2953.8	588.0 a (377.5)	93.7–1557.6	854.9 a (658.5)	259.8–3134.1	2197.3 b (1635.1)	672.6–7039.6	1210.0 ab (587.3)	743.5–2068.5
Dicaffeoylquinic acid derivative	13,666.3 b (5106.7)	5796.8–25,411.0 *	5209.6 a (1652.8)	1829.5–8635.3 *	9143.7 c (2813.5)	4444.9–17,185.1	8714.1 c (3020.2)	2429.9–13,267.3 *	5974.8 ac (655.3)	5134.3–6717.9
Phenolic acid I derivative	3411.3 ab (1488.4)	1050.0–9318.8	2971.1 a (1054.3)	867.5–5157.4	4052.3 b (1664.2)	1777.9–9153.9	3272.0 ab (830.5)	1850.4–4924.6	3875.0 ab (1067.2)	2948.5–5414.6 *
Phenolic acid II derivative	3539.3 b (1216.8)	1263.7–5728.5	2346.5 a (576.2)	1075.8–3514.6	3673.4 b (1346.6)	2095.4–7532.2	3201.1 b (1319.3)	1049.3–5706.5 *	3695.5 b (409.5)	3331.8–4265.6
Phenolic acid III derivative	2170.2 a (741.8)	593.2–3915.3 *	2765.6 ab (1019.6)	942.6–4848.4	3620.2 b (1930.7)	971.5–9347.3	1267.3 c (528.3)	659.1–2305.8	1895.0 abc (1152.2)	860.2–3020.1
Phenolic acid IV derivative	6606.5 b (4188.0)	1842.9–21,480.2 *	868.0 ac (770.0)	98.8–4683.0	1834.0 a (2323.0)	216.0–14,780.1	658.8 c (671.0)	66.2–2585.2	196.0 c (69.0)	131.1–278.8
Total	37,734.4 b (11,977.2)	16,200.3–61,935.5	19,429.9 a (4912.8)	7794.5–32,284.8 *	29,535.2 b (8382.3)	16,467.4–51,243.8	31,510.2 b (9895.2)	13,256.2–50,049.7 *	25,297.1 ab (2201.0)	23,301.7–28,300.0
ABTS	205.0 b (52.5)	83.3–305.0	154.4 a (42.2)	86.7–312.2	193.1 b (39.8)	122.2–289.4	187.4 ab (50.1)	115.6–301.7 *	131.9 a (9.5)	118.9–141.1

^1^ Number of sampled root accessions. ^2^ Values (mean ± standard deviation) followed by different letters within the row were significantly different between species at *p* ≤ 0.05, according to the Kruskal–Wallis test. * Significant differences among populations of species according to ANOVA at *p* ≤ 0.05; ** Significant differences among populations of species according to ANOVA at *p* ≤ 0.05.

**Table 3 plants-13-00132-t003:** Correlation matrices for the variables and the two principal components in the PCA1 and PCA2 models.

Variables	PCA1		PCA2	
	PC 1	PC 2	PC 1	PC 2
Neochlorogenic acid	0.72	−0.48	0.60	−0.48
Chlorogenic acid	0.79	−0.05	0.65	0.66
Cryptochlorogenic acid	0.82	−0.41	0.83	−0.11
4,5-dicaffeoylquinic acid	0.73	0.22	0.82	0.26
1,5-dicaffeoylquinic acid	0.48	−0.23	0.27	0.02
3,5-dicaffeoylquinic acid	0.70	0.41	0.77	0.50
3,4-dicaffeoylquinic acid	0.84	0.24	0.78	0.50
Dicaffeoylquinic acid derivative	0.83	−0.26	0.59	−0.64
Phenolic acid I derivative	0.47	0.58	0.55	−0.47
Phenolic acid II derivative	0.75	0.35	0.73	−0.34
Phenolic acid III derivative	0.08	0.75	0.21	−0.76
Phenolic acid IV derivative	0.44	−0.45	−0.01	−0.75

**Table 4 plants-13-00132-t004:** Pearson correlation coefficients between individual phenolic compounds and antioxidant activity, as assessed by ABTS in five *Solidago* spp. root extracts.

Variables	*S. virgaurea*	*S. canadensis*	*S. ×niederederi*	*S. gigantea*	*S. ×snarskisii*	Total
Neochlorogenic acid	0.52 **	−0.19	0.27	0.31	−0.69	0.43 **
Chlorogenic acid	0.49 **	0.04	0.41 *	0.46 *	−0.24	0.33 **
Cryptochlorogenic acid	0.68 **	−0.01	0.37 *	0.37	−0.17	0.51 **
4,5-dicaffeoylquinic acid	0.64 **	0.22	0.38 *	0.15	−0.23	0.40 **
1,5--dicaffeoylquinic acid	0.13	−0.10	−0.07	−0.29	−0.42	0.09
3,5-dicaffeoylquinic acid	0.51 **	0.27	0.41 *	0.31	−0.36	0.38 **
3,4-dicaffeoylquinic acid	0.70 **	0.29	0.50 **	0.30	−0.21	0.38 **
Dicaffeoylquinic acid derivative	0.74 **	0.37 *	0.42 *	0.28	0.25	0.63 **
Phenolic acid I derivative	0.67 **	0.45 *	0.46 *	0.07	0.55	0.49 **
Phenolic acid II derivative	0.70 **	0.44 *	0.61 **	0.07	0.10	0.57 **
Phenolic acid III derivative	0.53 **	0.40 *	0.36 *	−0.15	0.84	0.24 *
Phenolic acid IV derivative	0.2	−0.05	0.03	0.09	0.09	0.30 **
Total	0.8 **	0.46 *	0.65 **	0.36	0.35	0.70 **

Correlation significance level: * *p* < 0.05; ** *p* < 0.001.

## Data Availability

Data is contained within the article.

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
