# Peer review of "Comparative Analysis of Root Phenolic Profiles and Antioxidant Activity of Five Native and Invasive Solidago L. Species"

_plants, 2024, doi:10.3390/plants13010132_

Round 1
Reviewer 1 Report
Comments and Suggestions for Authors
This paper (plants-2781437) described the identification of differences in root phenolic profiles between five Solidago species to provide valuable information on their potential applications and the botanical origin of the raw material. In this study, it was used the roots of native S. virgaurea, two alien species: S. gigantea and S. canadensis, as well as their hybrids S. ×niederederi and S. ×snarskisii were sampled from mixed-species stands in nineteen sites of Lithuania. As a result, a complex of twelve phenolic acid and their derivatives was identified and quantified in methanol-water mixture root extracts using the HPLC-PDA with spectroscopic LC/MS system. Therefore, the reviewer recommended the publication of Plants after the revision as the following points out.
Major
1) The reviewer considered who and how was the plant species appraised. It should be described in detail. Since it is not accompanied by the results of genetic analysis, the reviewer considered that the appraisal by species morphology is very important.
2) The reviewer assumes PDA HPLC was used as the analytical instrument, and as long as using PDA, it would be better to analyze with max-plot (all wavelengths) for a more comprehensive analysis. As for the paper in general, why are the authr interested in phenolic compounds?
3) Related to the above question, is there a correlation between the antioxidant effects of phenolic compounds and "invasive" Solidago spp. If there is a correlation, a detailed description should be written.
4) The reviewer should describe what the loading plot of the principal component analysis looks like. If there are species-specific compounds, the results will be very interesting. The scaling in the principal component analysis should be well described.
Minor
1) Scientific names of plants are not correctly italicized. Especially after page 5, italicized plant names are not listed correctly. Please check again.
Reviewer 2 Report
Comments and Suggestions for Authors
The authors here presented their research regarding root phenolic profiles and antioxidant activities of five cloolected Solidago species.
The results is too simple without profound investigation and I am afraid the current manuscript is difficult to be further assessed in the current stage.
My concerns:
1. The samples is not sufficient and more samples from broader area should be collected.
2. It is not sufficient that only ABTS was available in assessment of anti-oxidative stress and at least one more methods should be used.
Author Response
"Please see the attachment.

Reviewer 3 Report
Comments and Suggestions for Authors
Dear Authors,
The Manuscript ID: plants-2781437, Titled “Comparative analysis of root phenolic profiles and antioxidant activity of five native and invasive Solidago species” is well-designed. It states the purpose of the review, the principal results and conclusions.
Based on the evaluation of its originality, significance of content, scientific soundness, and interest to readers, a major revision is recommended. Specific suggestions and comments are provided below.
The title is informative. It declares clearly the object of the article. However, I would suggest author citation of the plant genus to be added too.
The Abstract is factual and well-structured. Nevertheless, it should be added what kind of phenolic acids were identified. Additionally, author citation of the plant species to be written too. Also, the abstract should be a total of about 200 words maximum.
I should recommend a Graphical abstract to be presented.
The Introduction presents comprehensive data concerning the settled object. The literature survey is wide-ranging. Nevertheless, this section does not provide sufficient background and does not include all relevant references e.g., some previously achieved data on the antioxidant activity of Solidago species should be added too (Marksa, M.; Zymone, K.; Ivanauskas, L.; Radušienė, J.; Pukalskas, A.; Raudonė, L. Antioxidant profiles of leaves and inflorescences of native, invasive and hybrid Solidago species. Ind. Crop. Prod. 2020, 145, 112123. 635
https://doi.org/10.1016/j.indcrop.2020.112123)
Material and methods. The extraction procedure is not clear. There is missing data about the solvent and etc. and it should be written. Additionally, other tests and examinations for antioxidant inhibitory activity are needed, e.g., DPPH, FRAP, CUPRAC and etc. for proving the free radical scavenging activity.
Results and Discussion. Results are based on the received results. A good discussion is presented. However, based on the above-mentioned suggestions, additional examination should be done and the results should be edited.
The Conclusions are based on the founded data.
Round 2
Reviewer 1 Report
Comments and Suggestions for Authors
The revised manuscript was properly corrected, thus the reviewer recomended this paper to publish in Plants.
Author Response
Thank you very much for taking the time to review our manuscript.
Reviewer 2 Report
Comments and Suggestions for Authors
The revised manuscript is improved.
I recommend that minor reversion is needed before accept.
My comments:
1. In the discussion section, the authors should add explanation regarding the antioxidant effects on the basis of their findings of phenolic compounds.
Example papers are recommended to cite:
1). Du, B., Cheng, C., Chen, Y., Wu, J., Zhu, F., Yang, Y., & Peng, F. (2021). Phenolic profiles and antioxidant activities of exocarp, endocarp, and hypanthium of three pear cultivars grown in China. Journal of Food Bioactives, 14. https://doi.org/10.31665/JFB.2021.14269
2). Meng D, Zhang P, Zhang L, Wang H, Ho CT, Li S, Shahidi F, Zhao H. Detection of cellular redox reactions and antioxidant activity assays. Journal of Functional Foods. 2017;37:467-479
Author Response
Comparative analysis of root phenolic profiles and antioxidant activity of five native and invasive Solidago L. species
|
Response to Reviewer 2 Comments Thank you very much for taking the time to review our manuscript. Please find the response below and the corresponding corrections in the re-submitted manuscript. |
|
Comment 1: In the discussion section, the authors should add explanation regarding the antioxidant effects on the basis of their findings of phenolic compounds. Example papers are recommended to cite: Meng D, Zhang P, Zhang L, Wang H, Ho CT, Li S, Shahidi F, Zhao H. Detection of cellular redox reactions and antioxidant activity assays. Journal of Functional Foods. 2017;37:467-479 . Du, B.; Peng, F.; Cheng, C.; Chen, Y.; Wu, J., Zhu F.; Yang, Y. Phenolic profiles and antioxidant activities of exocarp, endocarp, and hypanthium of three pear cultivars grown in China. J. Food Bioact. 2021,14: 75–80. |
Response 1: Thanks to the Reviewer for the valuable comment. In the discussion we present some considerations that we believe are relevant to the purpose of our work.
Overall, the differences in antioxidant capacity confirmed that phenolic compounds contribute to the different of antioxidant capacity performance in different organs of goldenrod. The different scavenging activity was reported even in different parts of the Pyrus communis L. fruit, such as exocarp, endocarp and hypanthium, depending on the content of the main compounds in them [57]. Thus, the quantification of the individual phenolics is important for predicting the potential of antioxidant capacity in different parts of the plant and to reveal more a targeted use of plant materials. In this way, the high content of dicaffeoylquinic acid and phenolic acid I, II and IV derivatives apparently resulted in the highest antioxidant activity of S. virgaurea. Chlorogenic acid, as the main phenolic compound, in the phenolic profiles of both areal parts and roots of S. gigantea, can be considered as a marker of antioxidant activity of this species.
In this context, it was important to use the same radical scavenging capacity method to compare the relative antioxidant activity of different plant materials and to assess their variation trends. Thus, the ABTS assay, which is a well-known method for the determination of total antioxidant activity applicable to both lipophilic and hydrophilic com-pounds, was used as in previous of Solidago spp. studies to assess the radical scavenging response of phenolic compounds [43]. (lines: 373–389)
Reviewer 3 Report
Comments and Suggestions for Authors
Dear Authors,
Based on the comments and suggestions which were given, an enhancement of Manuscript ID: plants-2781437 is done.
In conclusion, based on the evaluation of the significance of content, scientific soundness, and interest to readers, after considering the improvements, I recommend the revised Manuscript ID: plants-2781437, to be published in Plants.
Author Response

(The authors gave the same response as above.)
